# Diagnosing fraudulent baseline data in clinical trials

**Michael A. Proschan**[1], **Pamela A. Shaw**[2]*

**1** Biostatistics Research Branch, National Institute of Allergy and Infectious Diseases, Bethesda, MD, United States of America, **2** Department of Biostatistics, Epidemiology, and Informatics, Philadelphia, PA, United States of America

* shawp@upenn.edu

## Abstract

The first table in many articles reporting results of a randomized clinical trial compares baseline factors across arms. Results that appear inconsistent with chance trigger suspicion, and in one case, accusation and confirmation of data falsification. We confirm theoretically results of simulation analyses showing that inconsistency with chance is extremely difficult to prove in the absence of any information about correlations between baseline covariates. We offer a reasonable diagnostic to trigger further investigation.

## 1 Introduction

In clinical trials, baseline variables are used to: 1) document that the trial recruited its target population, 2) summarize the natural history of the disease in the control arm, and 3) adjust the treatment effect for baseline differences in prognostic factors. Because baseline variables are measured before randomization, any differences between arms are attributable to chance. That is, the null hypothesis of no treatment effect should hold marginally for each baseline variable. For each continuous baseline variable compared using a continuous test statistic, the **marginal** distribution of its p-value should be uniform if the assumptions underlying the test (e.g., the data are normally distributed) are satisfied.

Clinical trialists sometimes go one step further and assume that p-values should behave like **independent** uniform deviates. Seeing appreciably more or fewer than the expected one in 20 statistically significant differences at $\alpha = 0.05$ arouses suspicion. In one case uncovered by Bolland et al. [1], that suspicion led to the accusation and later verification of data falsification. A randomized study in dogs uncovered by Calisle et al. [2] also appeared to show implausibly little variability of baseline covariates across arms. It and other publications by the same authors were retracted.

Betensky and Chiou [3] and Bland [4] use simulation to show that, in practice, p-values for baseline variables in clinical trials frequently do not behave like independent uniforms for several reasons: 1) the assumptions underlying a test may not be satisfied (e.g., skewed data do not fit the normality assumption), 2) the covariate may be binary, in which case even marginal uniformity of p-values does not hold exactly, and 3) many baseline covariates are correlated, so their p-values are also correlated. The authors conclude that interpretation of standard tests

**Data Availability Statement:** All relevant data were provided in the paper or were generated by the code available at https://github.com/PamelaShaw/FraudRCT.

**Funding:** The authors received no specific funding for this work.

**Competing interests:** No authors have competing interests.

of uniformity applied to p-values is problematic. A natural question is whether a legitimate case for data falsification can be made based solely on p-values reported in a baseline table (i.e., with no information presented on correlations between baseline covariates).

We propose a statistical test based on the sum of squared z-scores of baseline covariates that can be used to determine whether further investigation of fraud is warranted. This article complements the simulation results in [3–4] with theoretical results showing the difficulty of actually proving fraud. The problem is that the distribution of the test statistic depends critically on the correlation between z-scores comparing arms on baseline covariates. Naively treating these z-scores as independent rejects the null hypothesis of no fraud too often if there is any true correlation. On the other hand, using the worst case correlation matrix leads to an extremely conservative test that virtually never rejects the null hypothesis. We characterize correlation matrices that ought to be conservative, but not so conservative as to be useless.

## 2 Test statistic $L^2 = ||Z||^2$

### 2.1 Weighted combination of iid $\chi^2(1)$s

Let $P_i$ be the one-tailed p-value for testing whether treatment observations tend to be larger than control observations for the $i$th continuous baseline covariate, $i = 1, \ldots, k$. Assume that the corresponding test statistic has a continuous distribution and its underlying distributional assumptions are satisfied. In the absence of data falsification, the $P_i$ are dependent uniform (0, 1) random variables. Betensky and Chiou [3] evaluate the impact of correlation on chi-squared and Kolmogorov-Smirnov statistics of uniformity of the $P_i$. We consider instead a test specifically targeting too little variability of actual results from expected results, an indication of possible data falsification.

We begin by transforming the dependent uniforms $P_i$ to dependent standard normals $Z_i$ by $Z_i = \Phi^{-1}(1 - P_i)$, where $\Phi^{-1}$ is the inverse of the standard normal distribution function. Although the $Z_i$ need not have a multivariate normal distribution, they will be approximately multivariate normal if the test statistics are asymptotically sums of iid random variables and the sample size dwarfs the number of baseline covariates. Given that our intent is to show the difficulty of proving data falsification even under the best circumstances, we assume that the $Z_i$ are exactly multivariate normal with mean vector $(0, 0, \ldots, 0)$ and nonnegative-definite covariance matrix $\Sigma$ whose diagonal elements are 1. That is, the $Z_i$ are correlated (unless $\Sigma$ is the identity matrix) but marginally standard normal.

We will suspect data falsification if the $Z_i$ are too close to their expected value of 0; i.e., there is too much balance between arms. The sign of the $Z_i$ is not important, so we will be suspicious if $Z_i^2$ is very small for multiple baseline variables. A natural way to combine the $Z_i^2$ is through $L^2 = ||\underline{Z}||^2 = \sum_{i=1}^{k} Z_i^2$, the squared length of the vector $\underline{Z}$ of z-statistics for the $k$ baseline covariates. If the $Z_i$ were iid, we could determine whether $L^2$ is "too small" by comparing its value to the $\alpha$th percentile of a chi-squared distribution with $k$ degrees of freedom. But the $Z_i$ are dependent, so we must derive the distribution of $L^2$ under a marginal standard normal assumption, but with arbitrary correlation matrix $\Sigma$. We derive this distribution using standard results in linear algebra (see, for example, [5]) as follows.

### 2.2 Distribution of $L^2$

We can find an orthonormal basis $\underline{e}_1, \ldots, \underline{e}_k$ of eigenvectors of $\Sigma$ and form the orthogonal matrix $\Gamma$ whose columns are $\underline{e}_1, \ldots, \underline{e}_k$. Then $\Gamma^T \Sigma \Gamma = D$, a diagonal matrix whose diagonal entries $\lambda_1, \ldots, \lambda_k$ are the eigenvalues of $\Sigma$, which are all nonnegative real numbers because $\Sigma$ is

a nonnegative-definite, symmetric matrix. Therefore,

$$\Sigma = \Gamma D \Gamma^T.$$

Let $\Sigma^{1/2}$ denote the matrix $\Gamma D^{1/2} \Gamma^T$, where $D^{1/2}$ is the diagonal matrix whose diagonal elements are the square roots of those of $D$. Then $\underline{Z}$ has the same distribution as $\Sigma^{1/2}\underline{\delta}$, where $\underline{\delta}$ is a vector of $k$ iid standard normals, because cov $(\Sigma^{1/2}\underline{\delta}) = \Sigma^{1/2}I(\Sigma^{1/2})^T = (\Gamma D^{1/2}\Gamma^T)(\Gamma D^{1/2}\Gamma^T) = \Gamma D \Gamma^T = \Sigma$. It follows that

$$
\begin{aligned}
L^2 = \|\underline{Z}\|^2 \quad &= \underline{\delta}^T \sum \underline{\delta} = \underline{\delta}^T \Gamma D \Gamma^T \underline{\delta} \\
&= (\Gamma^T\underline{\delta})^T D(\Gamma^T\underline{\delta}) \\
&= (\underline{Z}*)^T D \underline{Z}*,
\end{aligned}
\tag{1}
$$

where $\underline{Z}^* = \Gamma^T\underline{\delta}$. Also, cov $(\Gamma^T\underline{\delta}) = \Gamma^T I \Gamma = I$, so the distribution of $\underline{Z}^*$ is that of $k$ iid standard normals. Eq (2) implies that

$$L^2 = \sum_{i=1}^{k}\lambda_i Z_i^{*2} \tag{2}$$

is a weighted sum of squares of iid standard normals. The weights $\lambda_i$ are the eigenvalues of $\Sigma$, which sum to $k$.

We interpret these eigenvalues in terms of variances of linear combinations of the $Z_i$ as follows. The vector $\underline{c}$ maximizing the variance of the linear combination $\underline{c} \cdot \underline{Z}$, subject to $\|\underline{c}\|^2 = 1$, is $\underline{c} = \underline{e}_{(k)}$, the eigenvector associated with the largest eigenvalue, $\lambda_{(k)}$, of $\Sigma$. We can view $\underline{c} \cdot \underline{Z}$ as the projection of $\underline{Z}$ onto the axis defined by $\underline{c}$ (Fig 1). The variance of $\underline{e}_{(k)} \cdot \underline{Z}$ is $\lambda_{(k)}$. The second largest eigenvalue $\underline{e}_{(k-1)}$ of $\Sigma$ is the maximum variance of linear combinations $\underline{c} \cdot \underline{Z}$ such that 1) $\|\underline{c}\|^2 = 1$, and 2) $\underline{c}$ is orthogonal to $\underline{e}_{(k)}$. The variance of $\underline{e}_{(k-1)} \cdot \underline{Z}$ is $\lambda_{(k-1)}$. Continuing in this fashion, the smallest eigenvalue $\underline{e}_{(1)}$ of $\Sigma$ is the maximum variance of linear combinations $\underline{c} \cdot \underline{Z}$ such that 1) $\|\underline{c}\|^2 = 1$, and 2) $\underline{c}$ is orthogonal to each of $\underline{e}_{(k)}, \underline{e}_{(k-1)}, \cdots, \underline{e}_{(2)}$. The variance of $\underline{e}_{(1)} \cdot \underline{Z}$ is $\lambda_{(1)}$. If there are a few very large eigenvalues and the rest are close to 0, the $Z_i$ are highly correlated. On the other hand, if the eigenvalues are all of similar size, the $Z_i$ are close to being uncorrelated.

**Summary**:

1. The distribution of $L^2$ under correlation matrix $\Sigma$ for $\underline{Z}$ is a weighted sum of iid chi-squared random variables with 1 degree of freedom.

2. The weights are the eigenvalues of $\Sigma$, which sum to $k$.

3. If all eigenvalues are 1, the $Z_i$ are iid and $L^2 \sim \chi_k^2$.

4. If $k-1$ eigenvalues are 0, the $Z_i$ are maximally correlated and $L^2 \sim k\chi_1^2$.

## 2.3 Peakedness as a function of $\underline{\lambda}$

We have characterized the distribution of the test statistic $L^2$ in the absence of fraud and under an assumed correlation matrix $\Sigma$. Next we investigate the peakedness of this distribution. If the distribution is peaked, then a small value of $L^2$ suggests possible fraud, as small values would be quite unlikely otherwise. On the other hand, if the distribution of $L^2$ is very dispersed, then small values of $L^2$ might be common even under the null hypothesis of no fraud.

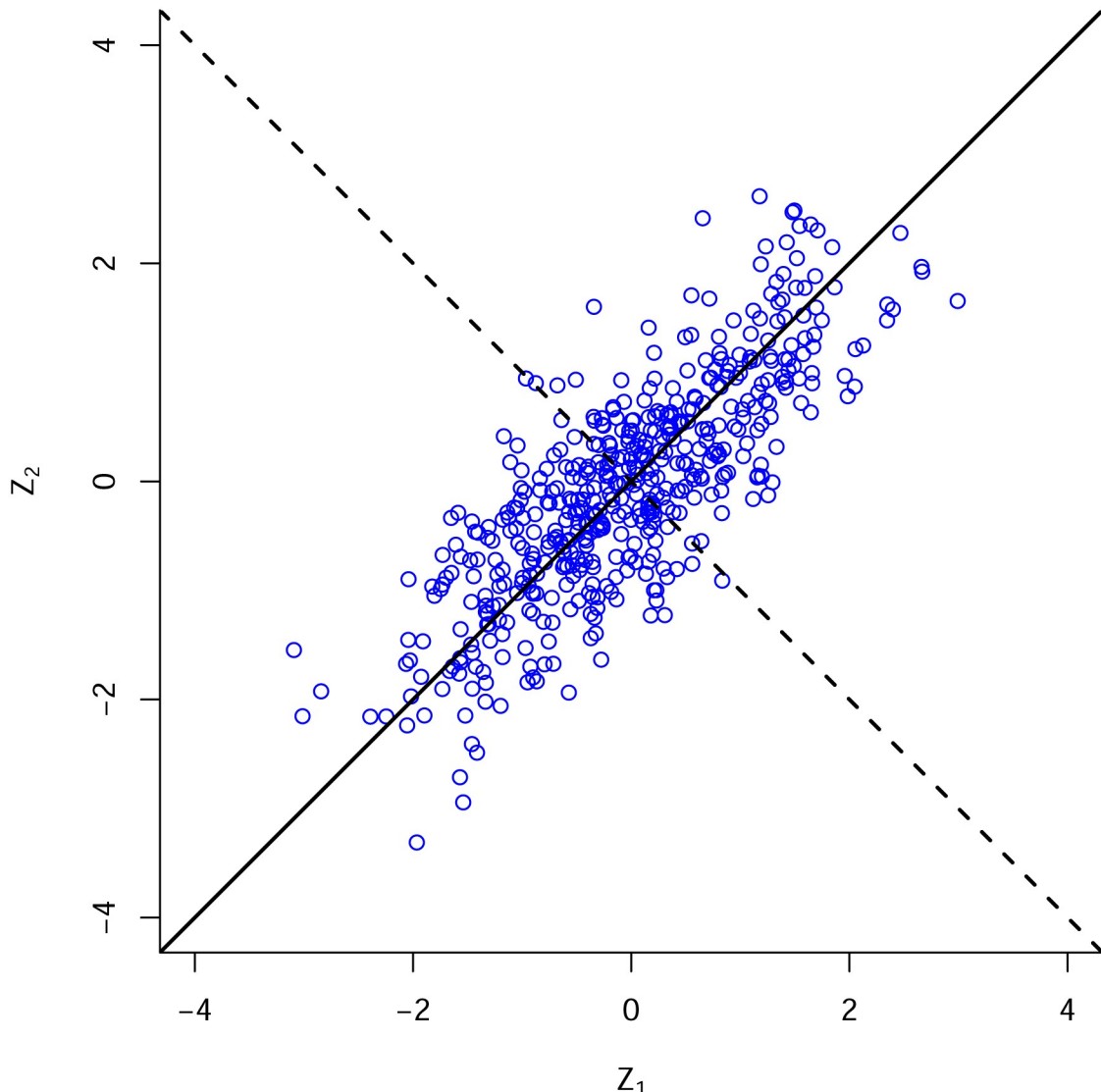

**Fig 1. Eigenvalues and eigenvectors when *k* = 2 illustrated with a large number of observations from a bivariate standard normal distribution with correlation *ρ* = 0.80. Observations vary most when projected onto the direction vector $\underline{e}_{\cdot 2} = (1/2^{1/2}, 1/2^{1/2})$ (solid line). The corresponding variance is $\lambda_{(2)}$, the larger of the two eigenvalues. The variance of values projected onto the orthogonal direction vector $\underline{e}_{\cdot 2} = (1/2^{1/2}, -1/2^{1/2})$ (dashed line) is the smaller eigenvalue, $\lambda_{(1)}$. The sum of the two eigenvalues is $\lambda_{(1)} + \lambda_{(2)} = 2$.**

It is critical, therefore, to determine limits on the peakedness of the null distribution to properly interpret evidence engendered by small values of $L^2$. There are different ways to quantify peakedness, but perhaps the simplest is based on the variance.

**2.3.1 Minimum and maximum variance.**   We consider next how the variance of $L^2$ depends on the eigenvalues of the correlation matrix of $\underline{Z}$. This will be important to evaluate how misleading it can be to treat p-values for baseline covariates as if they were independent.

The mean and variance of $L^2$ are

$$\mathrm{E}(L^2) \quad = \quad \sum_{i=1}^{k} \lambda_i \mathrm{E}\{\chi^2(1)\} = k \text{ and} \tag{3}$$

$$\mathrm{var}(L^2) \quad = \quad V = \sum_{i=1}^{k} \lambda_i^2 \mathrm{var}\{\chi^2(1)\} = 2\sum_{i=1}^{k} \lambda_i^2. \tag{4}$$

Thus, the mean of $L^2$ does not depend on the correlation matrix $\Sigma$ for the baseline z-scores, but the variance of $L^2$ does. For this reason, it is important to find the minimum and maximum values $V_{\min}$ and $V_{\max}$ of $\mathrm{var}(L^2)$ and determine which correlation matrices yield those extreme values. Because the $\lambda_i$ sum to $k$, $\bar{\lambda} = (1/k)\sum \lambda_i = 1$. Write $V$ as

$$\begin{aligned} V \quad &= 2\sum \lambda_i^2 - 2k\bar{\lambda}^2 + 2k\bar{\lambda}^2 = 2\sum_{i=1}^{k} (\lambda_i - \bar{\lambda})^2 + 2k \\ &= 2(k-1)s_\lambda^2 + sk, \end{aligned} \tag{5}$$

where $s_\lambda^2$ is the sample variance of the $\lambda_i$. It is clear that $V$ is minimized when $\lambda_i = \bar{\lambda} = 1$ for each $i$. In other words, the independence case, $\Sigma = I$, produces the smallest variance, $V_{\min} = 2k$, of $L^2$. In a sense, this produces the greatest peakedness for the null distribution of $L^2$. To see the serious implications of this fact, suppose that the observed value of $L^2$ is small. If we wrongly assume that the z-scores for baseline covariates are independent, then we will be using the minimum possible variance of $L^2$ to determine whether $L^2$ is implausibly small. Consequently, the observed $L^2$ value might be many assumed standard deviations away from its mean value of $k$. The resulting p-value will be tiny, and the level of evidence supporting data falsification will be greatly overstated if the true correlation matrix $\Sigma$ is far from the identity matrix corresponding to independent z-scores.

To avoid inflating the probability of erroneously suspecting data falsification, we could assume the correlation matrix $\Sigma$ yielding the largest variance of $L^2$. It can be shown that the $\underline{\lambda}$ maximizing the variance of $L^2$ assigns value $k$ to one of the $\lambda_i$ and 0 to the remaining $\lambda_i$s. In that case, $\mathrm{var}(L^2) = 2\sum \lambda_i^2 = 2k^2$. In summary, the smallest and largest values of $\mathrm{var}(L^2)$ are:

$$V_{\min} \quad = \quad 2k, \text{ corresponding to } \underline{\lambda} = (1, \ldots, 1) \tag{6}$$

$$V_{\max} \quad = \quad 2k^2, \text{ corresponding to } \underline{\lambda} = (0, 0, \ldots, k). \tag{7}$$

We will see that using $V_{\max}$ is almost always too conservative to be useful. Therefore, we want to select conservative values of $V$ that are not so conservative that they are useless. To see how to do this, notice that the vectors $(1, \ldots, 1)$ and $(0, 0, \ldots, k)$ in (6) and (7) are at opposite ends of a certain spectrum. Imagine two different communities, each with $k$ luxury cars divided among $k$ people. In one community, everyone has 1 luxury car, and in the other community, one person has all $k$ luxury cars. Vectors $(1, \ldots, 1)$ and $(0, 0, \ldots, k)$ correspond to these least and most polarized distributions. This concept can be formalized as follows. A vector $\underline{y}$ is said to *majorize* another vector $\underline{x}$, written $\underline{x} \prec \underline{y}$, if the ordered values $x_{(1)} \leq \ldots \leq x_{(k)}$ and $y_{(1)} \leq \ldots \leq y_{(k)}$ satisfy $\sum_{i=j}^{k} x_{(i)} \leq \sum_{i=j}^{k} y_{(i)}$ for $j = 1, \ldots, k$, with equality when $j = 1$. In other words, $\underline{y}$ is more polarized (the rich are richer and the poor are poorer) than $\underline{x}$. The smallest and largest vectors, in terms of majorization, with sum $k$ are $(y_{(1)}, \ldots, y_{(k)}) = (1, \ldots, 1)$ and $(0, \ldots, 0, k)$, respectively. The generalization of the ordering of variances in (6) and (7) is as follows.

**Theorem 2.1**. $V = \mathrm{var}(L^2)$ *increases as the vector* $\underline{\lambda}$ *of eigenvalues of* $\Sigma$ *becomes more polarized (i.e., increases in the majorization ordering).*

The proof follows from D.2 on page 101 of [6].

Therefore, computing the null distribution of $L^2$ assuming that $\underline{\lambda}$ is one of the larger (although not the largest possible) vectors in the majorization ordering should be conservative, but not prohibitively so.

Our treatment in this section implicitly assumed that the distribution of $L^2$ is approximately normally distributed for large $k$, which is reasonable for many $\underline{\lambda}$ because $L^2$ is a linear combination of iid chi-square random variables with 1 degree of freedom. However, for certain extreme vectors $\underline{\lambda}$ such as $(0, \ldots, k)$, $L^2$ is not normal. We would like to show, without invoking asymptotic normality, that the distribution of $L^2$ has fatter tails as $\underline{\lambda}$ becomes more polarized (increases in the majorization ordering). We defer discussion of this technical and difficult topic to the appendix.

## 2.4 Simple Σs allowing exact calculation

For any given critical value $C$, we can compute $P(L^2 \leq C)$ analytically without using a normal approximation for certain classes of correlation matrices $\Sigma$. Equivalently, we can think in terms of the eigenvalues of $\Sigma$, which, as we have seen, can be interpreted in terms of variances of projections of the $Z_i$ onto directions defined by its eigenvectors. Suppose the total variance is spread equally among a small number of directions. Then all but a few eigenvalues are 0, and the remainder all have the same value. For instance, with only 1 direction, all but one eigenvalue is 0, and the nonzero eigenvalue is $k$. This is the most extreme correlation matrix in which all $Z_i^2$ are identical. More generally, if all variability is focused equally in $j$ directions, then each of the $j$ nonzero eigenvalues has value $k/j$. In that case, expression (2) becomes $k(X_j/j)$, where $X_j$ has a chi-squared distribution with $j$ degrees of freedom. The probability of a type 1 error is

$$P\{k(X_j/j) \leq C\} = G_j(C/k), \tag{8}$$

where $G_j$ is the distribution function of $1/j$ times a chi-square random variable with $j$ degrees of freedom. The appendix shows that $G_j$ has fatter tails as $j$ decreases. Therefore, the distribution of $L^2$ has fatter tails if the total variability of $\underline{Z}$ is spread equally over a smaller number of directions.

Another relatively simple class of correlation matrices corresponds to the same correlation $\rho$ for all pairs of z-scores. It can be shown that all but one eigenvalue is $1-\rho$, and the remaining eigenvalue is $1 + (k - 1)\rho$. In that case, expression (2) can be written as

$$(1 - \rho)X_{k-1} + \{1 + (k - 1)\rho\}X_1, \tag{9}$$

where $X_{k-1}$ and $X_1$ are independent chi-squared random variables with $k - 1$ and 1 degree of freedom, respectively. Let $H_j$ and $h_j$ denote a chi-squared distribution and density function with $j$ degrees of freedom, $j = 1, \ldots, k$. From (9), the type 1 error rate is

$$P[(1 - \rho)X_{k-1} + \{1 + (k - 1)\rho\}X_1 \leq C] = \int H_{k-1}\left[\frac{C - \{1 + (k - 1)\rho\}x_1}{1 - \rho}\right]h_1(x_1). \tag{10}$$

Table 1 uses Eqs (8) and (10) to compute the inflation of the type 1 error rate when one erroneously assumes that the z-scores comparing baseline covariates across arms are independent, when the true $\Sigma$ is either equicorrelated or has all variability focused in a few directions. For the rows labeled by directions, the true $\Sigma$ corresponds to total variability of z-scores divided equally among 1, 2, or 3 directions. For rows labeled by $\rho$, the z-scores for baseline

**Table 1. Probability of suspecting fraud (that is, $L^2 \leq C$) when $C$ is determined assuming the $Z_i$ are independent.** The true $\Sigma$ either has all variability focused equally in 1, 2, or 3 directions, or each off-diagonal element has value $\rho$.

| Truth | $k = 1$ | $k = 5$ | $k = 10$ | $k = 25$ | $k = 100$ |
|---|---|---|---|---|---|
| 1 direction | .050 | .368 | .470 | .555 | .623 |
| 2 directions | — | .205 | .326 | .443 | .541 |
| 3 directions | — | .124 | .243 | .375 | .495 |
| $\rho = 0.00$ | .050 | .050 | .050 | .050 | .050 |
| $\rho = 0.25$ | .050 | .059 | .073 | .112 | .277 |
| $\rho = 0.50$ | .050 | .090 | .151 | .310 | .541 |
| $\rho = 0.75$ | .050 | .182 | .338 | .496 | .599 |

comparisons all have the same pairwise correlation $\rho$. For example, the "1 direction" row shows that if critical value $C$ is computed assuming the $Z_i$ are independent, but the truth is that all variability of the $Z_i$ is focused in only 1 direction, the actual type 1 error rate is 47.0 percent or 62.3 percent if $k = 10$ or $k = 100$, respectively. On the other hand, if the true $\Sigma$ has the same pairwise correlation $\rho = 0.50$ for all pairs, the true type 1 error rate is 15.1 percent or 54.1 percent for $k = 10$ or $k = 100$, respectively. The probability of falsely becoming suspicious increases as the $Z_i$ become more correlated.

On the other hand, if one assumes perfect correlation and sets the critical value using $V_{\max}$, the test becomes extraordinarily conservative. Table 2 shows that when $k = 25$, the actual type 1 error rate of the $V_{\max}$ test if the z-scores have common correlation $\rho = 0.75$ is $7.9 \times 10^{-20}$ instead of 0.05. In other words, if we want to protect against the most drastic correlation matrix $\Sigma_{ij} = 1$ for all $i$ and $j$, the test becomes incredibly conservative even if the true correlation matrix still has unrealistically high correlation. Likewise, even if the true correlation matrix has all variance focused in only 3 directions, the $V_{\max}$ test has ultraconservative type 1 error rate 0.0003. Remember that the $L^2$ test is being used as a diagnostic to see if further investigation is warranted. Further investigation would provide an estimate of $\Sigma$ that could be used to compute the true distribution of $L^2$, resulting in a much more accurate test. Thus, a reasonable option for the diagnostic test is to make a conservative assumption, such as that all correlations are 0.75 or all variability is focused equally in only 3 directions. This is almost guaranteed to overstate the degree of correlation in a real clinical trial.

**2.4.1 Example.** Fujii et al. [7] was a study randomizing 24 dogs to one of three doses of midazolam to evaluate the effect of midazolam on contractility of the diaphragm. Table 3 shows baseline means and standard deviations in each of the three arms for each of 8 continuous variables. There appears to be little variability across arms. We apply our test to dose

**Table 2. Probability of suspecting fraud (that is, $L^2 \leq C$) when $C$ is determined assuming $\Sigma_{ij} = 1$ for all $i, j$ (i.e., perfect correlation).** The true $\Sigma$ either has all variability focused equally in 1, 2, or 3 directions, or each off-diagonal element has value $\rho$.

| Truth | $k = 1$ | $k = 5$ | $k = 10$ | $k = 25$ | $k = 100$ |
|---|---|---|---|---|---|
| 1 direction | .050 | .050 | .050 | .050 | .050 |
| 2 directions | — | .004 | .004 | .004 | .004 |
| 3 directions | — | .0003 | .0003 | .0003 | .0003 |
| $\rho = 0.00$ | .050 | $2.8 \times 10^{-6}$ | $2.3 \times 10^{-11}$ | $2.3 \times 10^{-26}$ | $1.2 \times 10^{-100}$ |
| $\rho = 0.25$ | .050 | $3.5 \times 10^{-6}$ | $4.7 \times 10^{-11}$ | $2.8 \times 10^{-25}$ | $3.3 \times 10^{-95}$ |
| $\rho = 0.50$ | .050 | $6.4 \times 10^{-6}$ | $2.2 \times 10^{-10}$ | $2.6 \times 10^{-23}$ | $1.1 \times 10^{-86}$ |
| $\rho = 0.75$ | .050 | $2.2 \times 10^{-5}$ | $4.1 \times 10^{-9}$ | $7.9 \times 10^{-20}$ | $4.9 \times 10^{-72}$ |

Table 3. Fujii et al. [7] study in dogs uncovered by Carlisle et al. [2]. Shown are the eight continuous baseline covariates.

| Continuous var. | Dose Group 1 | Dose Group 2 | Dose Group 3 |
|---|---|---|---|
| mean (sd) | ($n = 8$) | ($n = 8$) | ($n = 8$) |
| HR (bpm) | 141 (15) | 143 (10) | 140 (12) |
| MAP (mm Hg) | 130 (15) | 132 (12) | 131 (11) |
| RAP (mm Hg) | 5 (2) | 5 (2) | 5 (2) |
| MPAP (mm Hg) | 12 (2) | 12 (2) | 12 (2) |
| PAOP (mm Hg) | 8 (2) | 8 (1) | 8 (2) |
| CO (L/min) | 2.2 (0.5) | 2.2 (0.4) | 2.3 (0.4) |
| Frequency (Hz) 20 | 15.5 (2) | 15.3 (1.8) | 15.4 (2.1) |
| Frequency (Hz) 100 | 21.1 (2) | 20.9 (2.2) | 20.9 (2.1) |

groups 1 and 2. For each variable, compute a one-tailed p-value $P_i$ using an unpaired t-statistic with alternative hypothesis that group 2 has a higher mean than group 1. Then convert each p-value to a z-score by $Z_i = \Phi^{-1}(1-P_i)$, and compute the test statistic $L^2 = \sum_{i=1}^{8} Z_i^2$. We find that $L^2 = 0.2556$. If we erroneously assume independence of baseline covariates, and therefore of $Z$ statistics, the p-value is $P(\chi_8^2 \leq 0.2556) \approx 10^{-5}$. This overstates the strength of evidence for fraud. On the other hand, assuming perfect correlation between z-scores for baseline covariates almost certainly understates the evidence for fraud. That p-value is $P(8\chi_1^2 \leq 0.2556) \approx 0.14$. We feel confident that a real randomized experiment would not result in all correlations being 0.9. Therefore, making the assumption of a common $\rho$ of 0.9 should still be highly conservative. The p-value using (10) with $\rho = 0.9$ is 0.0055. In other words, even under what we feel is an unrealistically large degree of correlation, namely 0.9 for all pairs, the evidence for fraud certainly seems sufficient to warrant further investigation.

## 3 Number of significant z-scores

We have focused on $L^2$ as a test statistic for detecting cheating, but other goodness of fit statistics such as those considered by Betensky and Chiou [3] and Bland [4] have similar behavior. A particularly simple statistic is the number $J$ of statistically significant z-scores. We might be suspicious if the number of continuous baseline covariates is large and none result in a statistically significant difference between arms. Suppose these z-scores are equicorrelated with non-negative correlation $\rho$. Then $Z_1, \ldots, Z_k$ have the same distribution as $X + \epsilon_1, \ldots, X + \epsilon_k$, where $X$ and the $\epsilon_i$ are mutually independent normal random variables with zero means and variances $\sigma_X^2 = \rho$, $\sigma_\epsilon^2 = 1 - \rho$. Let $z_\alpha$ satisfy $1 - \Phi(z_\alpha) = \alpha$. Given $X = x$, the indicators $I(Z_i > z_\alpha)$ are iid Bernoulli $(p)$ random variables, where

$$\begin{aligned} p &= p(x) = P(Z > z_\alpha | X = x) = P(x + \epsilon > z_\alpha) \\ &= 1 - \Phi\left\{\frac{z_\alpha - x}{\sqrt{1-\rho}}\right\} \end{aligned} \tag{11}$$

The specific distribution function $F(p)$ for the random variable $P = p(X)$ is unimportant. The important fact is that $P$ has mean $\alpha$, as the following calculation shows:

$$E(P) = E\{P(Z > z_\alpha | X)\} = P(Z > z_\alpha) = \alpha.$$

Accordingly, the distribution of $J$ is a mixed binomial:

$$P(J = j) = \int \binom{k}{j} p^j (1 - p)^{k-i} f(p) dp, \qquad (12)$$

where $f(p)$ is the density function corresponding to distribution function $F(p)$. Note that $\int pf(p)dp = \alpha$.

Under independence, the number of significant $Z_i$ has an ordinary binomial distribution with parameters $k$ and $\alpha$. let $J$ and $J'$ denote random variables from the mixed binomial (12) and the unmixed binomial bin$(k, \alpha)$. To see that extreme results are more likely for $J$ than for $J'$, note that by Jensen's inequality, for $k > 1$,

$$\begin{aligned} P(J = 0) &= \mathrm{E}\{P(J = 0|P)\} = \mathrm{E}\{(1 - P)^k\} \\ &> \{1 - \mathrm{E}(P)\}^k = (1 - \alpha)^k = P(J' = 0). \end{aligned} \qquad (13)$$

More generally, Shaked [8] has shown that a mixed binomial has fatter tails than an ordinary binomial with the same mean. This explains the common phenomenon of observing what appear to be too few or too many statistically significant baseline differences in clinical trials. Therefore, if one falsely assumes that the $Z_i$ are independent, the chance of falsely suspecting fraud will be inflated.

## 4 Discussion

We have proposed a diagnostic for detecting suspiciously low between-arm variability in baseline covariates in clinical trials. The test statistic $L^2$, the squared length of the vector $\underline{Z}$ of z-scores of balance in baseline covariates, has a distribution that depends on the correlation matrix $\Sigma$ of $\underline{Z}$ only through its eigenvalues. We confirm analytically for $L^2$ what is demonstrated through simulation in [3–4] for similar goodness of fit tests applied to baseline covariates in clinical trials when no information about correlations is available. Assuming independence between covariates (and, therefore, between the $Z_i$) results in an unacceptably high probability of falsely suspecting fraud. In fact, the distribution of $L^2$ has thinnest tails when one falsely assumes that z-scores for baseline covariates are independent, and fattest tails when one assumes the most extreme possible correlation. We draw two conclusions: 1) one should never conclude fraud solely because $L^2$ is unusually small under the independence assumption and 2) to feel confident that the $Z_i$ are too small to have occurred by chance, $L^2$ must be unusually small even assuming unrealistically high correlation. Assuming perfect correlation produces a test that virtually never triggers further investigation. Therefore, we suggest using a practical upper bound on the correlation matrix such as all correlations equal to 0.75 or all variability focused equally in only 3 directions.

The final verdict will almost always be based on the totality of evidence. The case made by Bolland et al. [1] was based on numerous trials by the same authors that contained warning signs such as suspiciously fast enrollment and few deaths and dropouts despite recruiting older patients with substantial comorbidity. Our test statistic is a useful diagnostic that can be used in conjunction with other evidence to bolster the case for data falsification.

## A Appendix: Fat-tailed distributions

The argument in Section 2.3 that $L^2$ should have fatter tails as $\underline{\lambda}$ becomes more polarized was based on approximate normality of $L^2$ for large $k$. But if $\underline{\lambda} = (0, \ldots, 0, k)$, $L^2$ has the distribution of $k$ times a chi-squared variate with 1 degree of freedom, which is not normal. This

section addresses whether $L^2$ has fatter tails as $\underline{\lambda}$ becomes more polarized even without the approximate normality assumption.

The first problem lies in defining "fatter tails.– This is easy for normal distributions or other distributions symmetric about their mean: the same mean and larger variance implies fatter tails. For asymmetric distributions such as those of linear combinations of chi-squared random variables, we must use a different definition. One possibility is the following.

**Definition A.1**. *Distribution function $F_2$ has fatter tails than distribution function $F_1$, denoted by $F_1 \overset{f}{<} F_2$ or $X_1 \overset{f}{<} X_2$, if there exists a number $x^*$ such that $F_2(x) \geq F_1(x)$ for $x \leq x^*$, and $F_2(x) \leq F_1(x)$ for $x > x^*$.*

In other words, $F_1 \overset{f}{<} F_2$ if the left tail $F_2(x)$ is at least as large as the left tail $F_1(x)$ for all $x \leq x^*$, and the right tail $1-F_2(x)$ is at least as large as the right tail $1-F_1(x)$ for all $x > x^*$. Another way of expressing this fact is that if $F_1-F_2$ has any sign changes, then there is exactly one, and the sequence of signs is $-$, $+$ as $x$ increases.

Bock, et al. [9] conjectured, but did not prove the following.

**Conjecture A.1**. (*Bock et al.* [9]) *if $Z_1^*, \ldots, Z_k^*$ are iid standard normals and $\underline{\lambda}_1 \prec \underline{\lambda}_2$, then*

$$\underline{\lambda}_1 \cdot \underline{Z}^{*2} \overset{f}{<} \underline{\lambda}_2 \cdot \underline{Z}^{*2}.$$

Theorem 1 of Roosta-Khorasani and Szekely [10] is closely related, but it shows that large values are more likely for $\underline{\lambda}_2 \cdot \underline{Z}^{*2}$ than for $\underline{\lambda}_1 \cdot \underline{Z}^{*2}$. We are interested in the opposite tail, namely that very small values are more likely for $\underline{\lambda}_2 \cdot \underline{Z}^{*2}$ than for $\underline{\lambda}_1 \cdot \underline{Z}^{*2}$.

Although we have been unable to prove Conjecture A.1 in complete generality, empirical evidence and heuristic arguments support its veracity. For example, we investigated the $k = 2$ case using an extensive grid of possible values of $\underline{\lambda}_1$ and $\underline{\lambda}_2$ and computing the distributions of $\underline{\lambda}_1 \cdot \underline{Z}^{*2}$ and $\underline{\lambda}_2 \cdot \underline{Z}^{*2}$ through numerical integration. For $k = 3$, we repeatedly generated random vectors $\underline{\lambda}_1$ and $\underline{\lambda}_2$ from a simplex in a way that $\underline{\lambda}_1 \prec \underline{\lambda}_2$, and used simulation to compute the distributions of $\underline{\lambda}_1 \cdot \underline{Z}^{*2}$ and $\underline{\lambda}_2 \cdot \underline{Z}^{*2}$. Further details are available from the authors.

One special case of Conjecture A.1 is when all of the variability of $\underline{Z}$ is concentrated equally in each of $j$ directions. In that case, the distribution of $L^2$ is $(k/j)\underline{v} \cdot \underline{Z}^{*2}$, where $\underline{v}$ contains $j$ ones and $k-j$ zeroes. Since $\underline{v} \cdot \underline{Z}^{*2}$ has a chi-squared distribution with $j$ degrees of freedom, $L^2$ is $k(\chi_j^2/j)$, where $\chi_j^2$ denotes a chi-squared random variable with $j$ degrees of freedom. Thus, Conjecture A.1 says that $\chi_j^2/j$ has fatter tails as $j$ diminishes. Although we are unable to prove Conjecture A.1, we prove this special case at the end of the appendix.

**Theorem A.1** $\chi_j^2/j \overset{f}{<} \chi_i^2/i$ *for integers $i, j$, $i < j$.*

Another important special case of Conjecture A.1 is when all pairs $(Z_i, Z_j)$ have the same correlation $\rho \geq 0$. In that case, the eigenvalue vector is $(1-\rho, \ldots, 1-\rho, 1 + (k-1)\rho)$, which increases in the majorization ordering as $\rho$ increases. The conjecture implies that $L^2$ has fatter tails as $\rho$ increases. Eq (9) shows that $L^2 = \tau(\chi_1^2/1) + (1 - \tau)(\chi_k^2/k)$, where $\tau = \{1 + (k-1)\rho\}/k$. Therefore, Conjecture A.1 applied to the special case of equicorrelated $Z_i$s is equivalent to $\tau(\chi_1^2/1) + (1 - \tau)(\chi_k^2/k)$ having fatter tails as $\tau$ increases from $1/k$ to 1.

It should be noted that one could define fatness of tails of a distribution function in ways other than Definition A.1. For example, suppose that $E\{\psi(X)\} \leq E\{\psi(Y)\}$ for every convex function $\psi$ such that the expectations exist. Then not only is the variance of $X$ no greater than that of $Y$, but the same is true for the fourth central moment, the sixth central moment, etc. This is one way of formulating the idea that the distribution function of $Y$ has fatter tails than that of $X$.

It is very easy to prove, using the alternative definition above, that assuming the $Z_i$ are independent produces the thinnest tailed distribution. This is demonstrated in Theorem A.2.

**Theorem A.2** *Let $Z_1^*, \ldots, Z_k^*$ be iid standard normals and $Y_{\underline{\lambda}} = \underline{\lambda} \cdot (Z_1^{*2}, \ldots, Z_k^{*2})$, where $\sum_{i=1}^{k} \lambda_i = k$. Then for any convex function $\psi$ such that $\mathrm{E}\{\psi(Y_{\underline{\lambda}})\}$ is finite for all $\underline{\lambda}$, $\mathrm{E}\{\psi(Y_{\underline{\lambda}})\}$ is largest when $\underline{\lambda} = (0, \ldots, 0, k)$.*

This follows from the definition of convex function: for any $u_1, \ldots, u_k$ and nonnegative $w_1, \ldots, w_k$ with $\Sigma w_i = 1$, $\psi(\Sigma w_i u_i) \leq \Sigma w_i \psi(u_i)$. Set $w_i = \lambda_i/k$, $i = 1, \ldots, k$. Then

$$
\begin{aligned}
\mathrm{E}\left\{\psi\left(\sum_{i=1}^{k} \lambda_i Z_i^{*2}\right)\right\} &= \mathrm{E}\left\{\psi\left(\sum_{i=1}^{k} w_i k Z_i^{*2}\right)\right\} \\
&\leq \sum_{i=1}^{k} w_i \mathrm{E}\{\psi(k Z_i^{*2})\} = \sum_{i=1}^{k} w_i \mathrm{E}\{\psi(k Z_k^{*2})\} \\
&= \mathrm{E}\{\psi(k Z_k^{*2})\} \sum_{i=1}^{k} w_i = \mathrm{E}\{\psi(k Z_k^{*2})\} \\
&= \mathrm{E}[\psi\{(0, \ldots, 0, k) \cdot (Z_1^{*2}, \ldots, Z_k^{*2})\}],
\end{aligned}
\tag{14}
$$

completing the proof.

**Proof of Theorem A.1**

Let $f_j(x)$ be the density of $\underline{\lambda}^{(j)} \cdot \underline{Z}^{*2}$, where $\{(k-j)/k\}\underline{\lambda}^{(j)} = (0, \ldots, 0, 1, \ldots, 1)$ has $k{-}j$ zeroes followed by $j$ ones. The distribution of $\{(k-j)/k\}\underline{\lambda}^{(j)} \cdot \underline{Z}^{*2}$ is chi-squared with $j$ degrees of freedom. It follows that

$$
h(x) = \frac{f_{j-1}(x)}{f_j(x)} = \left\{\frac{\sqrt{2}\,\Gamma(j/2)\left(\frac{j-1}{j}\right)^{j/2}}{\Gamma(j/2 - 1/2)\sqrt{\frac{j-1}{k}}}\right\}\left\{\frac{\exp\left(\frac{x}{2k}\right)}{\sqrt{x}}\right\}
\tag{15}
$$

The derivative of $g(x) = \exp(x/2k)/x^{1/2}$ is negative for $0 \leq x < k$ and positive for $x > k$. Thus, $g(x)$ decreases for $x < k$ and increases for $x > k$. Moreover, $g(x)$ has a limit of $+\infty$ as either $x \downarrow 0$ or $x \to \infty$. These facts also hold for $h(x)$, which is just a positive constant times $g(x)$. It follows that the number $m_1$ of $x$ such that $h(x) = 1$ is either 0, 1, or 2. But $m_1$ cannot be 0 or 1 because that would imply that $f_{j-1}(x) > f_j(x)$ for all $x$ or for all but one $x$, contradicting the fact that $f_{j-1}(x)$ and $f_j(x)$ both integrate to 1. Therefore, $h(x) = 1$ for exactly two values, $x = x_1$ and $x = x_2$, $x_1 < x_2$.

Let $F_{j-1}(x)$ and $F_j(x)$ be the distribution functions corresponding to the densities $f_{j-1}(x)$ and $f_j(x)$. Because $f_{j-1}(x) > f_j(x)$ for $x < x_1$ and $x > x_2$, $F_{j-1}(x) > F_j(x)$ for $x < x_1$ and $F_{j-1}(x) < F_j(x)$ for $x > x_2$. Because $F_{j-1}$ and $F_j$ are continuous, there must be a point $x^*$ for which $F_{j-1}(x^*) = F_j(x^*)$. Necessarily, $x_1 < x^* < x_2$. But $f_{j-1}(x) < f_j(x)$ for $x \in (x_1, x_2)$, so if $F_{j-1}(x^*) = F_j(x^*)$, then $F_{(j-1)}(x) < F_j(x)$ for all $x \in (x^*, x_2)$. But we have already established that $F_{j-1}(x) < F_j(x)$ for $x > x_2$, so $F_{j-1}(x) < F_j(x)$ for $x > x^*$. Putting these facts together, we have established that

$$
F_{j-1}(x) - F_j(x) \begin{cases} > 0 & \text{if } x < x^*, \\ = 0 & \text{if } x = x^*, \\ < 0 & \text{if } x > x^*. \end{cases}
$$

This completes the proof.

## Author Contributions

**Methodology:** Michael A. Proschan, Pamela A. Shaw.

**Writing – original draft:** Michael A. Proschan.

**Writing – review & editing:** Pamela A. Shaw.

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
