## [Decision Letter · Decision Letter 0]

28 Nov 2019

PONE-D-19-23183

Detecting Fraudulent Baseline Data in Clinical Trials

PLOS ONE

Dear Dr Proschan,

Thank you for submitting your manuscript to PLOS ONE. After careful consideration, we feel that it has merit but does not fully meet PLOS ONE’s publication criteria as it currently stands. Therefore, we invite you to submit a revised version of the manuscript that addresses the points raised during the review process.

We would appreciate receiving your revised manuscript by Jan 12 2020 11:59PM. To enhance the reproducibility of your results, we recommend that if applicable you deposit your laboratory protocols in protocols.io, where a protocol can be assigned its own identifier (DOI) such that it can be cited independently in the future. For instructions see: http://journals.plos.org/plosone/s/submission-guidelines#loc-laboratory-protocols

We look forward to receiving your revised manuscript.

Kind regards,

Vance W. Berger, PhD

Academic Editor

PLOS ONE

Journal Requirements:

Additional Editor Comments:

I do understand that the entire goal was to confirm, mathematically, what was already demonstrated via simulation. Nevertheless, some readers might benefit from more intuition spread throughout the entire manuscript.

Reviewers' comments:

Reviewer's Responses to Questions

**Comments to the Author**

1. Is the manuscript technically sound, and do the data support the conclusions?

Reviewer #1: Yes

Reviewer #2: Yes

2. Has the statistical analysis been performed appropriately and rigorously? 

Reviewer #1: Yes

Reviewer #2: Yes

3. Have the authors made all data underlying the findings in their manuscript fully available?

Reviewer #1: Yes

Reviewer #2: No

4. Is the manuscript presented in an intelligible fashion and written in standard English?

Reviewer #1: Yes

Reviewer #2: Yes

5. Review Comments to the Author

Reviewer #1: This is a very important paper pointing out the limitations of using the opposite of a common approach of criticizing a study - that is, the balance of baseline covariates between randomized groups. My biggest lament is that because of all the theorems and theoretical nuances, this will not be widely read and it should be.

I wish the authors had used a larger study as opposed to an animal study with 8 dogs per group.

I have no criticisms of the statistical methods, they seem clear and correct. There are some minor issues, for example in discussing Figure 1, they authors state "...a huge number" -huge is of course a relative number, but a bit more precision or another adjective would be better. On page 9 of the manuscript (15 of the submission) the discussion of majorization and the entire paragraph might be a bit of obfuscation by statistics.

The Bolland article that utilized the lack of variability as one of its key bits of evidence is really not exactly the same issue that this paper is addressing, since Bolland et. al. used 33 studies. While one cannot routinely evaluate someone's entire portfolio to look for too little variation, the caution not to base it on a single study is both important and essential given the consequences in this day and age of often being convicted just by accusation.

Depending on the presumed audience, some of the theory and conjectures might be modified. Nevertheless, the paper is well done, educational and supported nicely by theory - so any modifications it would seem to be based on what audience the authors want to reach.

Reviewer #2: This is a mathematical paper that shows the underlying mathematics behind what is reasonably obvious and has been shown by simulation previously.

The issue is whether the distribution of P values for between treatment comparisons of many variables recorded at baseline in a randomised trial should show a uniform distribution. This will be so if all the variables are statistically independent and the assumptions behind the statistical test used to make the between treatment comparison are true. When the variables are not independent but correlated, there will not be a uniform distribution. This paper has the mathematical foundations for what seems clear, and this reviewer does not have the mathematical skills to be able to assess if all the mathematics is correct, but there are no obvious flaws and the overall results are as might be expected, but the theoretical justification could be seen as helpful.

There are various questions to be raised in relation to its possible publication in PLoS-One.

1 Is this the right journal for a fairly mathematical paper which will be inaccessible to most readers of PLoS-One? My view is that it is more suitable for a mathematical journal, but that is an editorial decision.

2 Is the title a reflection of the paper’s contents? The paper effectively suggests that detection of fraudulent data based on baseline variables is not possible.

3 What does this paper add, for the ordinary reader interested in the topic, to what has been written in a very much more accessible way by Bland [ref 4 in this manuscript- PLoS ONE. 2013;8(10): e76010.], and by Betensky & Chiou in ref 3 also[PLoS ONE. 2017;12(9):e0184531.]?

4 Bland, in examining his simulations, used a correlation of 0.5 between all the baseline variables, and Betensky & Chiou examined a range of correlations. The model that assumes a fixed level for all the correlations between variables is itself unrealistic. Real randomised trials have varying levels of the correlation between the variables and some will be close to independent. This paper examines various possibilities but a great deal of the emphasis seems to be based on quite high values of the between variable correlation and it being similar among all variables. It would be very much more helpful to have used a large amount of real trial data to examine actual correlations between baseline variables.

5 The paragraph on page 15 of the manuscript states (with some of the symbols altered)-

“On the other hand, if there is perfect correlation between z-scores for baseline covariates, the p-value is P =0.14. This yields insufficient evidence to conclude fraud. In reality, perfect correlation between z-scores is extremely unlikely. If we are confident that the maximum correlation is 0.9, the p-value using (9) is 0.0055. Thus, if we are confident that the maximum correlation of z-scores is 0.9, the evidence for fraud is still fairly strong. On the other hand, if we believe z-scores could have correlation 0.99, the evidence is not nearly as strong: p = 0.126.”

To me this has two problems. Firstly, I very much doubt if anyone who investigates possible fraud would “conclude” there was fraud on the basis of such statistical tests. At the most it would be a marker for possible fraud that would justify detailed investigation or would be one part of a variety of indicators that, when taken together, provided evidence of misconduct that would need to be refuted by an author. They of course, if the data were legitimate, could produce the exact data from which the correlation matrix could be calculated. Secondly, I know of no real data from RCTs in which correlations among all the baseline variables were anything like 0.9, and most would be less than 0.5. perhaps my experience is incorrect, but some evidence on actual values would be helpful. The authors should respond on these issues.

6 Is the final conclusion “Only the most blatant forms of cheating can be detected on the basis solely of p-values for baseline comparisons” justified only if one assumes unrealistic values for all the correlations? As noted above, examinations of the data using L2 or other methods can be helpful in looking at suspect data, and it is a wider range than just “the most blatant” that can show such problems. It is undoubtedly true that caution must be taken, and just unthinkingly applying such tests may suggest evidence of misconduct when there is none, but at the same time, it is also important that misconduct is investigated and there is a danger that this paper could be used to prevent any statistical examination of data to detect misconduct. I doubt that is the intention of the authors, but one must be aware of unintended consequences.

6. PLOS authors have the option to publish the peer review history of their article (what does this mean?). If published, this will include your full peer review and any attached files.

Reviewer #1: No

Reviewer #2: No

---

## [Author Response · Author response to Decision Letter 0]

30 Mar 2020

We thank the reviewers for their helpful comments. We have made several changes and feel the paper is much improved. In particular, we have increased the general accessibility of the paper by adding more detailed explanations of results and moving more of the technical material to an appendix.

Summary of Changes

1. To make the paper easier to read, we have:

(a) moved most of the technical material to the appendix,

(b) added a short summary at the end of Section 2.2 of results abouteigenvalues and their implications on the distribution of the test statistic L2, and

(c) eliminated any discussion of Schur functions.

2. We now emphasize that L2 is intended as a diagnostic test to trigger further investigation, not as definitive proof of fraud. We also changed the title of the paper to highlight diagnosing, rather than proving, fraud. 

Reviewer 1

1. This is a very important paper pointing out the limitations of using the opposite of a common approach of criticizing a study – that is, the balance of baseline covariates between randomized groups. My biggest lament is that because of all the theorems and theoretical nuances, this will not be widely read and it should be.

Thank you for your comment. As noted in item 1 of the above summary of changes, we have taken steps to make the paper more accessible to a more general audience.

2. I wish the authors had used a larger study as opposed to an animal study with 8 dogs per group. We used this as an example because it was cited by Carlisle et al. (2015) as showing implausibly little variability and indeed it does! There is much greater balance between arms than would be seen in an actual experiment.

3. I have no criticisms of the statistical methods, they seem clear and correct. There are some minor issues, for example in discussing Figure 1, they authors state ”...a huge number” –huge is of course a relative number, but a bit more precision or another adjective would be better. On page 9 of the manuscript (15 of the submission) the discussion of majorization and the entire paragraph might be a bit of obfuscation by statistics. We have changed “huge” to “large”. We have also reworded the discussion of majorization and better explained its relevance. As noted in item 1c of the above summary of changes, we have also eliminated any mention of Schur functions.

4. The Bolland article that utilized the lack of variability as one of its key bits of evidence is really not exactly the same issue that this paper is addressing, since Bolland et. al. used 33 studies. While one cannot routinely evaluate someone’s entire portfolio to look for too little variation, the caution not to base it on a single study is both important and essential given the consequences in this day and age of often being convicted just by accusation. You are correct that the Bolland article is different from ours in that it used 33 studies. The last paragraph of the Discussion section of the revision now includes this important difference in the context of the need to consider the totality of evidence before reaching a conclusion of fraud. We point out some of the other factors that Bolland et al. noted were suspicious.

5. Depending on the presumed audience, some of the theory and conjectures might be modified. Nevertheless, the paper is well done, educational and supported nicely by theory – so any modifications it would seem to be based on what audience the authors want to reach. Both reviewers pointed out that the mathematical content may be off-putting to some readers. We understand this criticism, and we have attempted to reword the paper to make it more accessible (see item 1 of the above summary of changes). We believe that the paper is now easier to read and digest the main messages while still containing the relevant underlying mathematics in the appendix. 

Reviewer 2

1. This is a mathematical paper that shows the underlying mathematics behind what is reasonably obvious and has been shown by simulation previously. We agree that many of the points in this article have been made before; however, we provide new mathematical insights and a practical test to detect fraudulent data. We also have moved more technical details to the appendix to make it more widely accessible.

2. The issue is whether the distribution of P values for between treatment comparisons of many variables recorded at baseline in a randomised trial should show a uniform distribution. This will be so if all the variables are statistically independent and the assumptions behind the statistical test used to make the between treatment comparison are true. When the variables are not independent but correlated, there will not be a uniform distribution. This paper has the mathematical foundations for what seems clear, and this reviewer does not have the mathematical skills to be able to assess if all the mathematics is correct, but there are no obvious flaws and the overall results are as might be expected, but the theoretical justification could be seen as helpful. Thank you. We agree.

3. There are various questions to be raised in relation to its possible publication in PLoS-One.

(a) Is this the right journal for a fairly mathematical paper which will be inaccessible to most readers of PLoS-One? My view is that it is more suitable for a mathematical journal, but that is an editorial decision. PLoS ONE actually contains a mix of different types of papers, some of which are fairly mathematical, which is consistent with the stated scope of the journal to be inclusive of a wide selection of subject areas that span science, medicine and engineering. Given that the Betensky and Chiou (2017) paper was in PLoS ONE, and our intent was to buttress their conclusions with theoretical justifications, we feel that this is the appropriate journal. We also believe that the revisions we have made make the paper more accessible to a more general audience.

(b) Is the title a reflection of the paper’s contents? The paper effectively suggests that detection of fraudulent data based

on baseline variables is not possible. Actually, detection of frauduent data is possible, but proving fraud is very difficult. We have changed the title to reflect our intention for L2 to be used as a diagnostic tool to trigger further investigation. The last paragraph of the Discussion section in the revision emphasizes that the totality of the evidence should be used to decide whether fraud has been proven.

(c) What does this paper add, for the ordinary reader interested in the topic, to what has been written in a very much more accessible way by Bland [ref 4 in this manuscriptPLoS ONE. 2013;8(10): e76010.], and by Betensky & Chiou in ref 3 also[PLoS ONE. 2017;12(9):e0184531.]? We believe that our diagnostic statistic L2 is particularly well suited to detect a common sign of possible fraud–too little variability. Additionally, we believe that supporting simulation results by theoretical arguments always strengthens those results. Only finitely many simulations can be undertaken, but a proof can cover an uncountably infinite number of settings. We are sympathetic to your concern that the presentation should be made more accommodating to less mathematically oriented readers, and we have attempted to do so (see item 1 in the above summary of changes).

(d) Bland, in examining his simulations, used a correlation of 0.5 between all the baseline variables, and Betensky & Chiou examined a range of correlations. The model that assumes a fixed level for all the correlations between variables is itself unrealistic. Real randomised trials have varying levels of the correlation between the variables and some will be close to independent. This paper examines various possibilities but a great deal of the emphasis seems to be based on quite high values of the between variable correlation and it being similar among all variables. It would be very much more helpful to have used a large amount of real trial data to examine actual correlations between baseline variables. We agree that the settings considered by most authors, including ourselves, are oversimplified. The problem is that we do not have a compendium of correlation matrices from baseline tables of different trials. No baseline table ever reports correlations between variables, and that is a real problem when one tries to gauge the strength of evidence against chance as an explanation for too much balance of baseline variables. Our paper reduces the dimension of the correlation matrix from k2 to k by showing that only the k eigenvalues matter. We further simplify matters by consider-

ing whether the total variability is concentrated in a small number of orthogonal directions, a sign of high correlation.

(e) The paragraph on page 15 of the manuscript states (with some of the symbols altered)– “On the other hand, if there is perfect correlation between z-scores for baseline covariates, the p-value is P =0.14. This yields insufficient evidence to conclude fraud. In reality, perfect correlation between z-scores is extremely unlikely. If we are confident that the maximum correlation is 0.9, the p-value using (9) is 0.0055. Thus, if we are confident that the maximum correlation of z-scores is 0.9, the evidence for fraud is still fairly strong. On the other hand, if we believe z-scores could have correlation 0.99, the evidence is not nearly as strong: p = 0.126.” To me this has two problems. Firstly, I very much doubt if anyone who investigates possible fraud would conclude there was fraud on the basis of such statistical tests. At the most it would be a marker for possible fraud that would justify detailed investigation or would be one part of a variety of indicators that, when taken together, provided evidence of misconduct that would need to be refuted by an author. They of course, if the data were legitimate, could produce the exact data from which the correlation matrix could be calculated. Secondly, I know of no real data from RCTs in which correlations among all the baseline variables were anything like 0.9, and most would be less than 0.5. perhaps my experience is incorrect, but some evidence on actual values would be helpful. The authors should respond on these issues. You have an excellent point that no one should conclude fraud on the basis of a baseline table with no information about correlations. Unfortunately, this point is not fully appreciated by the applied community, as witnessed by incorrect statistical methods that assumed independence among variables that were clearly not independent. We now make clear that any of these methods should be the basis for further investigation, not accusation. See also the last paragraph of the Discussion section in the revision that emphasizes the need to consider the totality of evidence before concluding fraud. Regarding the point about real data from clinical trials not having correlations as high as 0.9, we agree that we know of no clinical trials in which all correlations are 0.90, but we have certainly seen very high individual correlations. For example, the correlation between systolic and diastolic blood pressure is quite high. In lifestyle intervention trials, such as weight loss trials, there are high correlations between physical activity by total and all the separate components: walking, light activity, and moderate+vigorous. The important point is that if the p-value from the L2 test is very low even when an unrealistically high degree of correlation is assumed, the evidence of fraud is strong. We have added text to the example to more clearly make this point.

Is the final conclusion Only the most blatant forms of cheating can be detected on the basis solely of p-values for baseline comparisons justified only if one assumes unrealistic values for all the correlations? As noted above, examinations of the data using L2 or other methods can be helpful in looking at suspect data, and it is a wider range than just the most blatant that can show such problems. It is undoubtedly true that caution must be taken, and just unthinkingly applying such tests may suggest evidence of misconduct when there is none, but at the same time, it is also important that misconduct is investigated and there is a danger that this paper could be used to prevent any statistical examination of data to detect misconduct. I doubt that is the intention of the authors, but one must be aware of unintended consequences. This is a good point. We have changed the wording of the paper somewhat to emphasize that L2 is a useful diagnostic tool to justify further investigation

---

## [Editor Report · Decision Letter 1]

1 Sep 2020

Diagnosing Fraudulent Baseline Data in Clinical Trials

PONE-D-19-23183R1

Dear Dr. Shaw,

We’re pleased to inform you that your manuscript has been judged scientifically suitable for publication and will be formally accepted for publication once it meets all outstanding technical requirements.

Kind regards,

Vance Berger

Academic Editor

PLOS ONE
---

## [Editor Report · Acceptance letter]

14 Sep 2020

PONE-D-19-23183R1 

Diagnosing Fraudulent Baseline Data in Clinical Trials 

Dear Dr. Shaw:

I'm pleased to inform you that your manuscript has been deemed suitable for publication in PLOS ONE. Congratulations! Your manuscript is now with our production department. 

Kind regards, 

on behalf of

Dr. Vance Berger 

Academic Editor

PLOS ONE